# Clustering of Environmental Parameters and the Risk of Acute Ischaemic Stroke

**DOI:** 10.3390/ijerph20064979

**Published:** 2023-03-11

**Authors:** Geraldine P. Y. Koo, Huili Zheng, Joel C. L. Aik, Benjamin Y. Q. Tan, Vijay K. Sharma, Ching Hui Sia, Marcus E. H. Ong, Andrew F. W. Ho

**Affiliations:** 1Ministry of Health Holdings, Singapore 099253, Singapore; 2National Registry of Diseases Officer, Health Promotion Board, Singapore 168937, Singapore; 3Environmental Epidemiology and Toxicology Division, Environmental Health Institute, National Environment Agency, Singapore 228231, Singapore; 4Pre-Hospital & Emergency Research Center, Duke-NUS Medical School, Singapore 169857, Singapore; 5Division of Neurology, Department of Medicine, National University Hospital, Singapore 119074, Singapore; 6Yong Loo Lin School of Medicine, National University of Singapore, Singapore 119228, Singapore; 7Department of Cardiology, National University Heart Centre Singapore, Singapore 119074, Singapore; 8Health Services & Systems Research, Duke-NUS Medical School, Singapore 169857, Singapore; 9Department of Emergency Medicine, Singapore General Hospital, Singapore 169608, Singapore; 10Centre of Population Health Research and Implementation, SingHealth Regional Health System, Singapore 168753, Singapore; 11Saw Swee Hock School of Public Health, National University of Singapore, Singapore 117549, Singapore

**Keywords:** ischaemic stroke, cerebrovascular disease, air pollution, environmental epidemiology, clustering, haze

## Abstract

Acute ischaemic stroke (AIS) risk on days with similar environmental profiles remains unknown. We investigated the association between clusters of days with similar environmental parameters and AIS incidence in Singapore. We grouped calendar days from 2010 to 2015 with similar rainfall, temperature, wind speed, and Pollutant Standards Index (PSI) using k-means clustering. Three distinct clusters were formed ‘Cluster 1’ containing high wind speed, ‘Cluster 2’ having high rainfall, and ‘Cluster 3’ having high temperatures and PSI. We aggregated the number of AIS episodes over the same period with the clusters and analysed their association using a conditional Poisson regression in a time-stratified case-crossover design. Comparing the three clusters, Cluster 3 had the highest AIS occurrence (IRR 1.09; 95% confidence interval (CI) 1.05–1.13), with no significant difference between Clusters 1 and 2. Subgroup analyses in Cluster 3 showed that AIS risk was amplified in the elderly (≥65 years old), non-smokers, and those without a history of ischaemic heart disease/atrial fibrillation/vascular heart disease/peripheral vascular disease. In conclusion, we found that AIS incidence may be higher on days with higher temperatures and PSI. These findings have important public health implications for AIS prevention and health services delivery during at-risk days, such as during the seasonal transboundary haze.

## 1. Introduction

Stroke is a leading cause of disability and death worldwide, making it an important public health priority [1]. Emerging evidence shows that air pollution and meteorological parameters are significant modifiable risk factors for acute ischaemic stroke (AIS) [2,3]. Mechanistically, air pollutants, especially particulate matter, have been postulated to affect systemic inflammatory and coagulation pathways, oxidative stress, endothelial vasodilatory regulation, and epigenetics of atheromatous plaque formation and thrombosis [4,5,6]. Meteorological parameters such as temperature, on the other hand, may be involved in the risk of thrombotic events causing AIS, including cold-induced increased sympathetic activity, peripheral vasoconstriction, increased platelet count, blood pressure, blood viscosity, heat-induced dehydration, haemoconcentration, and increased cholesterol levels [2,7].

In Southeast Asia, the annual transboundary haze phenomenon occurs due to seasonal uncontrolled agricultural practices in Indonesia. Large areas of land are cleared by fire for vegetation, causing multiple forest and peat land fires [8]. Such biomass burning releases large amounts of greenhouse gases and particulate matter that alters local air quality. During this transboundary haze period, quantities of both gaseous and particulate pollutants, including PM2.5 and PM10, increase to greater than during non-haze periods, often rising to hazardous levels that exceed the air quality guidelines set by the WHO [9,10,11]. Airway deposition of particulate matter in the respiratory system was also found to be greater during the haze-to-non-haze period, with harmful downstream health effects [10,12]. Hence, this seasonal increase in air pollution poses an enormous public health and economic burden to the region [12,13]. Traditionally, the health effects of air pollution and meteorological parameters have been studied individually, while adjusting for each other as confounders. In general, air pollution levels were higher in Asia compared with Europe and America, while stroke risks were greater in Asian countries, particularly East Asia [14,15,16,17]. However, when comparing across regions, pooled hazard ratios for stroke were, contrariwise, highest in Europe with great heterogeneity in Asia [18]. Studies on meteorological parameters, mostly on temperature and AIS risk, also yielded contrasting results with great heterogeneity in the studied variables [19,20,21,22]. Nonetheless, most studies were conducted in temperate countries still with a paucity of evidence coming from the equatorial climate zone and lower- and middle-income countries (LMIC).

In recent years, there has been growing evidence that both air pollution and meteorological parameters may exert modifying effects on each other, although findings remain inconclusive [23,24,25]. Furthermore, with rising air pollution levels and the evolving threat of climate change, especially the potential health impact of air pollution and climate change, there is increasing relevance to studying such associations [26]. Local adaptive behaviour and other complex biometeorological confounders have also been shown to affect such associations [20,22]. Hence, it may be pertinent to study these associations based on geographical regions.

This study aims to identify clusters of days with distinct meteorological characteristics and air quality conditions and determine their association with AIS risk.

## 2. Materials and Methods

### 2.1. Setting

Singapore is a densely populated, small urban city-state with a population of 5.4 million and over 730 km^2^ of land space. It is situated at the tip of the Malaysian Peninsula, near the equator and temperatures are largely uniform throughout the year with an abundance of humidity and rainfall. However, it experiences two monsoon seasons—the Southwest Monsoon from June to September and the Northeast Monsoon from December to March—with the interim dry season from September to October, frequently coinciding with the El Nino–Southern Oscillation and positive Indian Ocean Dipole. This causes dry weather conditions that usually intensify the severity of haze [9].

Cerebrovascular disease, including stroke, is the fourth leading cause of death in Singapore and accounted for 6.1% of the deaths in 2021 [27]. With a rapidly ageing population, the incidence of stroke is expected to rise in Singapore [28].

### 2.2. Study Population

Data on AIS incidence was obtained from the Singapore Stroke Registry. Stroke cases are notified to the registry from medical claims made to the Ministry of Health, inpatient discharge summaries, and death data from the Ministry of Home Affairs. The International Classification of Diseases (ICD) diagnosis codes, ICD-9 codes 430 to 437 (excluding 432.1 and 435), and ICD-10 codes from I60 to I69 (excluding I62.0 and I62.1), are used to identify stroke cases. The registry coordinators confirmed the diagnosis of stroke by reviewing the patient’s medical records, before extracting individual-level clinical data relevant to the stroke episode. Recurrent stroke is defined as a separate event if it occurred 28 days after the preceding stroke, consistent with the Monitoring Trends and Determinants in Cardiovascular Disease, World Health Organization criteria. The type of stroke is based on computed tomography scans and/or magnetic resonance imaging data, aligned with the doctor’s diagnoses documented in the medical records.

The outcome variable in our study was daily AIS occurrence in Singapore from 2010 to 2015. We excluded patients with AIS that occurred during their hospital stay as these patients were already admitted and not exposed to the environmental conditions prior to out-of-hospital AIS. The event date was taken as the date of presentation to the Emergency Department (ED), as it was the earliest available date that was unambiguous to recall bias. Individual-level AIS data were aggregated into a format that summarised the number of AIS on each event date for analysis.

### 2.3. Environmental Data

The Pollutant Standards Index (PSI) is used in Singapore to report air quality. Six pollutants are used for its computation: fine particulate matter (PM2.5), particulate matter (PM10), carbon monoxide (CO), ozone (O_3_), nitrogen dioxide (NO_2_), and sulphur dioxide (SO_2_). A sub-index is calculated for each pollutant, from a segmented linear function, which transforms ambient concentration into a scale from 0 to 500. The maximum of the six sub-indices is PSI. Measurements of the six pollutants are obtained from more than 20 telemetric air quality monitoring stations distributed across the island [29]. Five meteorological stations around Singapore measure wind speed, rainfall, and temperature. The exposure variables in our study included data from 2010 to 2015 on mean wind speed, total rainfall, daily mean temperature, and 24 h mean PSI in Singapore. The daily environment data was merged with the aggregated AIS data by event date for analysis. Air quality and meteorological data were retrieved from the publicly available National Environment Agency and Meteorological Service Singapore websites, respectively.

### 2.4. Statistical Analysis

Within the study period of 2010 to 2015, mean wind speed, total rainfall, daily mean temperature, and PSI were used to classify the 2191 calendar days. Employing a two-step clustering, days were separated into clusters using the log-likelihood distance measure. We used the Akaike and Bayesian information criteria to obtain an optimum number of clusters. The two-step clustering, instead of hierarchical clustering and the k-means approach, was chosen as it does not require a predefined number of clusters and both ordinal and numeric variables can be used. Clusters formed using environmental data ordinal variables were more balanced, in terms of sample size, compared to numeric variables. Mean wind speed (0 to <7, ≥7 to <10, >10 km/h), total rainfall (0, >0 to <2, ≥2 mm), daily mean temperature (<27.5, >−27.5 to <28.5, ≥28.5 degree Celsius) and mean PSI (0 to ≤50, >50 to ≤100, >100) were categorised into ordinal variables. Three clusters were formed within the study period using the two-step clustering with days with the missing environmental data excluded. The Kruskal–Wallis rank test was used to compare the three clusters to see if there were any significant differences in the patients’ demographics, environmental parameters, and AIS clinical characteristics.

To examine the association between each cluster and the incidence of AIS, a time-stratified case-crossover approach was used. A case is a day with at least one AIS. Control periods were the same day, month, and year of the AIS onset date of the corresponding case and each case served as its own control. Overdispersion and autocorrelation were accounted for in the conditional Poisson regression model used to compare the incidence rate ratio (IRR) of AIS across the clusters of days [30]. We assessed the relationship between AIS incidence between clusters and subgroup analyses. STATA SE 13 was used for statistical analyses.

## 3. Results

### 3.1. Environmental Clusters

To demonstrate an association between the environmental factors, we used univariable and multivariable linear regression with PSI as outcome variation, temperature as an independent variable, and wind speed and rainfall as covariates. A positive correlation was found for temperature and PSI (Pearson correlation = 0.32), with a low correlation between PSI and wind speed (Pearson correlation = 0.088) and rainfall (Pearson correlation = −0.096) (refer to Figure 1 of our acute myocardial infarction (AMI) study [31]). In the 2191 days from 2010 to 2015, 686 days (31%) were classified into Cluster 1, 1029 days (47%) into Cluster 2, and 2467 days (21%) into Cluster 3; the remaining nine days were not classified as at least one of the environmental parameters was missing (refer to Table 1 of our AMI study for characteristics of each Cluster [31]). Generally, a higher daily average wind speed was seen in Cluster 1 (median 10.4 km/h; interquartile range [IQR] 8.4–12.2), higher daily total rainfall in Cluster 2 (median 2.8 mm; IQR 0.4–13.6), and higher daily average 24 h average PSI (median 59.2; IQR 54.8–70.0) and temperature (median 28.6 degrees C; IQR 27.8–29.0) in Cluster 3.

### 3.2. AIS

There were 29,384 episodes of AIS in the 2191 days between 2010 and 2015. Table 1 shows the characteristics of the study population in the three clusters. The median age of the AIS patients was 68.4 years (IQR 58.7–78.3). The three clusters had similar cardiovascular risk profiles in terms of the history of stroke/transient ischaemic attack (TIA) (~48%), ischaemic heart disease (IHD)/atrial fibrillation (AF)/valvular heart disease (VHD)/peripheral vascular disease (PVD) (~51%), diabetes mellitus (~56%), hypertension (~83%), hyperlipidaemia (~75%), and smoking (~41%). The survival rate at discharge was also similar across the clusters (~94%).

### 3.3. Association of Environmental Clusters with AIS Incidence

The incidence risk ratio (IRR) of AIS for Clusters 2 and 3, with Cluster 1 as a reference, is shown in Table 2. The incidence of AIS was higher in Cluster 3 (IRR 1.09, 95% confidence interval [CI] 1.05–1.13), compared to Cluster 1, with no significant difference in AIS incidence between Clusters 1 and 2 (IRR 0.98, 95% CI 0.95–1.00).

The higher IRR of AIS in Cluster 3 remained among all subgroups of patients, except Malays, Indians, those with a history of a previous stroke/TIA, and those without a history of hypertension, where the IRR did not differ significantly from Cluster 1. A lower IRR of AIS in Cluster 2 was observed among those aged 65 years and older, Chinese, with a history of IHD/AF/VHD/PVD, without a history of diabetes, with a history of hypertension, and with a history of hyperlipidaemia, where the IRR was lower than Cluster 1.

## 4. Discussion

In our time-stratified case-crossover analysis of AIS incidence from a population-based stroke registry, we found that days with higher ambient temperatures and PSI appeared to be associated with short-term increased risk of AIS. This is, to our knowledge, the first study in Southeast Asia, which demonstrated that clusters of days, according to their environmental characteristics, are associated with AIS risk.

A novel finding in our study was that AIS risk was higher on days with higher temperatures and PSI. We found only one other study that also employed cluster analyses to study stroke outcomes, although their findings were contrasting to ours—a recent Italian study found that days with low temperature and high ozone and days with moderately high temperatures and low pollutants were associated with increased stroke risk when compared to days with high temperatures and high ozone and days with low temperatures and low ozone levels [32]. A few studies that found associations between air pollution and stroke outcomes discovered stronger associations on colder days [33,34,35,36], while others on warmer days [35,37,38,39]. Although inconclusive, these findings show a possible modulating effect of weather on air pollution and vice versa on stroke outcomes. Studies that found a heating effect on air pollution and stroke risk suggested that time spent outside during the warmer weather led to greater air pollution exposure. Other possible mechanistic effects on this phenomenon could be due to similar pathophysiological pathways of air pollution exposure and heat on atheromatous plaque formation and rupture, exerting synergistic effects leading to AIS occurrence [35,37,39]. Studies that found a cold effect were mostly Chinese studies and they hypothesised that various factors such as decreased rainfall, stagnant air from less wind, and increased coal emissions during cold weather increased air pollution levels and exposure [33,36]. Of note, our other study on AMI also found a possible heat effect on air pollution and AMI occurrence [31]. Given similar pathophysiological pathways of atheromatous plaque formation in AIS and AMI, it further validates a possible heat effect on air pollution exposure and AIS occurrence.

Interestingly, in our subgroup analyses, we found that the AIS risk in Cluster 3 was amplified in non-smokers and those without a history of IHD/AF/VHD/PVD. Smoking is an established risk factor for AIS. Our findings suggest a possible tolerance effect from smoking [40]. Another reason could be a reduced residual confounding effect from a strong risk factor, such as smoking on AIS risk and air pollution exposure. A plausible biological explanation could be that similar oxidative stress and inflammation pathways of smoking and air pollution exposure for AIS resulted in no further enhanced effects from air pollution exposure in existing smokers [40,41,42,43]. We found it more difficult to justify our findings for IHD/AF/VHD/PVD. It could be that different risk factors, exposed to different environmental conditions, may exert differing effects on outcomes. In the Japanese Standard Stroke Registry Study (JSSRS), they found that the incidence of non-cardioembolic and lacunar stroke was highest in summer while that of cardioembolic stroke was highest in winter. They hypothesised that dehydration during hot summers led to hyper-viscosity and enhanced platelet aggregation, while sympathetic activation during cold winters led to higher blood pressure and atrial fibrillation. These mechanisms were involved in the differing pathogenesis of atherothrombotic and lacunar stroke and cardioembolic stroke respectively [44]. In our study, we studied AIS as a whole and grouped the risk factors IHD/AF/VHD/PVD together. This might make it difficult to predict the individual interactions of stroke subtypes and risk factors and their effect on stroke outcomes, possibly accounting for our findings.

In Cluster 3, we also found that the AIS risk was amplified in elderly patients (≥65 years old), although we did not find any sex-related modifications. Postulated influences of sex and age include occupational exposure, a sex-hormone-related and age-related capacity, and susceptibility to the thermoregulatory, and physiological response to temperature changes [45,46]. However, the evidence is mixed. Some studies found no sex or age-related modifications [47,48], others found greater AIS risks in males and the elderly [49], and others in females with cold exposure [50,51]. Given the wide variety of findings on subgroup modification, such associations remain controversial. 

Our study did not find significant associations between days with higher windspeed or higher rainfall and AIS risk. There is little evidence of the association between AIS risk and less extensively studied meteorological parameters such as rainfall and wind speed. In a study in Katowice, Poland, the authors found that in winter, high humidity and wind speed together with low temperature and sunshine increased the incidence of AIS, while in autumn, high humidity, and precipitation with low wind speed and sunshine decreases the risk of AIS [52]. Another systemic analysis of population-based studies found no association between atmospheric pressure and humidity with stroke admissions [53]. Overall, the interplay between environmental parameters and stroke outcomes is too complex and multifactorial, which makes drawing concrete conclusions difficult. More studies, especially from Southeast Asia and low- and middle-income countries (LMIC), where there seems to be a paucity of such studies, would be needed to further elucidate associations between environmental parameters and AIS risks.

There are a few limitations in our study. Firstly, this is an ecological study, thus, we were unable to prove a causal relationship between the environmental parameters and stroke outcomes studied. Secondly, we were unable to fully control for all confounding factors, including behavioural changes, socioeconomic status, and ability to undertake mitigating actions in relation to weather and air pollution. These might have potential risk-modifying effects on stroke outcomes. Thirdly, some studies have shown a possible delayed and cumulative effect of environmental exposure and AIS risk. However, in our study, due to the unfeasibility of lag analyses on different clusters, we only investigated short to intermediate environmental exposure and AIS risk. Lastly, due to the limited availability of air pollution data, we were unable to examine the independent effect of constituent air pollutants on AIS outcomes. By using an air quality index such as PSI, differing effects of each pollutant on cerebrovascular outcomes could have been mitigated. Nonetheless, PSI is a readily assessable and easily interpretable tool for air pollution for the general public. It remains a useful applicator for health education and policy implementation in the real world.

Our study adds a Southeast Asian context to the growing literature on the health impact of environmental parameters on AIS. Finding a possible association between characteristic daily weather and air pollution profiles and increased AIS risk, allows for improved evidence-based resource allocation. This includes potential cross-agency collaborations, with measures including mobile app alerts, healthcare warnings, and advisories, allocation of resources such as N95 masks, and emergency services deployment for high-risk exposure days.

## 5. Conclusions

We found a short-term association between clusters of environmental factors and AIS incidence. These findings have public health implications for AIS prevention and emergency health services delivery.

## Figures and Tables

**Table 1 ijerph-20-04979-t001:** AIS characteristics of the clusters (*n* = 29,384 AIS episodes).

	All (*n* = 29,384)	Cluster 1 (*n* = 9116)	Cluster 2 (*n* = 13,330)	Cluster 3 (*n* = 6825)	*p*-Value *
**Age, median (IQR)**	68.4 (58.7–78.3)	68.7 (58.7–78.5)	68.0 (58.4–77.9)	68.6 (59.4–78.7)	<0.001
**Male, *n* (%)**	17,165 (58.4)	5320 (58.4)	7812 (58.6)	3967 (58.1)	0.800
**Ethnicity, *n* (%)**					
Chinese	22,136 (75.3)	6940 (76.1)	9913 (74.4)	5187 (76.0)	0.056
Malay	4731 (16.1)	1429 (15.7)	2220 (16.7)	1071 (15.7)	
Indian	2021 (6.9)	603 (6.6)	964 (7.2)	449 (6.6)	
**History of stroke/TIA, *n* (%)**	7883 (48.2)	2490 (48.5)	3568 (48.2)	1795 (48.0)	0.867
**History of IHD/AF/VHD/PVD, *n* (%)**	9328 (51.7)	2949 (52.6)	4143 (51.2)	2197 (51.2)	0.223
**History of diabetes, *n* (%)**	11,474 (56.8)	3511 (56.5)	5259 (57.2)	2659 (56.3)	0.503
**History of hypertension, *n* (%)**	22,744 (82.9)	7109 (83.2)	10,241 (82.3)	5312 (83.6)	0.075
**History of hyperlipidaemia, *n* (%)**	18,490 (75.3)	5783 (75.7)	8241 (74.7)	4393 (76.0)	0.133
**Current/former smoker, *n* (%)**	11,673 (41.0)	3621 (41.0)	5367 (41.6)	2637 (39.7)	0.043
**Survived to hospital discharge, *n* (%)**	27,641 (94.1)	8548 (93.8)	12,557 (94.2)	6431 (94.2)	0.334

* Kruskal–Wallis test was used to obtain the *p*-values. IQR: interquartile range; TIA: transient ischaemic attack; IHD: ischaemic heart disease; AF: atrial fibrillation; VHD: valvular heart disease; PVD: peripheral vascular disease.

**Table 2 ijerph-20-04979-t002:** Incidence risk ratios (IRR) of AIS in Cluster 2 (high rainfall) and Cluster 3 (high temperature and PSI) compared to Cluster 1 (high wind speed).

	Cluster 1	Cluster 2	Cluster 3
**Number of AIS episodes**	9116	13,330	6825
**Average number of AIS per day**	13	13	15
**Entire cohort**	1.00	0.98 (0.95–1.00)	**1.09 (1.05–1.13) ^b,c^**
Subgroups			
**Age**			
<65 years	1.00	1.01 (0.97–1.05)	**1.06 (1.01–1.12) ^b,c^**
≥65 years	1.00	**0.95 (0.92–0.99) ^a^**	**1.12 (1.07–1.17) ^b,c^**
**Gender**			
Male	1.00	0.98 (0.94–1.02)	**1.08 (1.04–1.14) ^b,c^**
Female	1.00	0.97 (0.93–1.01)	**1.10 (1.05–1.16) ^b,c^**
**Ethnicity**			
Chinese	1.00	**0.95 (0.92–0.99) ^a^**	**1.09 (1.05–1.14) ^b,c^**
Malay	1.00	1.00 (0.95–1.06)	1.05 (0.98–1.12)
Indian	1.00	1.04 (0.98–1.10)	1.03 (0.96–1.10)
**History of stroke/TIA**			
Yes	1.00	0.96 (0.91–1.01)	1.05 (0.99–1.12)
No	1.00	0.98 (0.93–1.03)	**1.07 (1.01–1.14) ^b,c^**
**History of IHD/AF/VHD/PVD**			
Yes	1.00	**0.92 (0.88–0.97) ^a^**	**1.07 (1.02–1.14) ^b,c^**
No	1.00	1.02 (0.97–1.07)	**1.15 (1.09–1.22) ^b,c^**
**History of diabetes**			
Yes	1.00	1.00 (0.96–1.04)	**1.10 (1.05–1.16) ^b,c^**
No	1.00	**0.98 (0.94–1.03) ^a^**	**1.12 (1.06–1.19) ^b,c^**
**History of hypertension**			
Yes	1.00	**0.96 (0.93–0.99) ^a^**	**1.09 (1.04–1.13) ^b,c^**
No	1.00	1.01 (0.96–1.07)	1.04 (0.98–1.11)
**History of hyperlipidaemia**			
Yes	1.00	**0.95 (0.92–0.99) ^a^**	**1.10 (1.05–1.15) ^b,c^**
No	1.00	1.00 (0.95–1.06)	**1.07 (1.00–1.13) ^b,c^**
**Current/former smoker**			
Yes	1.00	0.98 (0.94–1.03)	**1.05 (1.00–1.11) ^b,c^**
No	1.00	0.96 (0.93–1.00)	**1.12 (1.07–1.17) ^b,c^**

^a^ Significant difference in IRR between Cluster 1 and Cluster 2 (*p* < 0.05). ^b^ Significant difference in IRR between Cluster 1 and Cluster 3 (*p* < 0.05). ^c^ Significant difference in IRR between Cluster 2 and Cluster 3 (*p* < 0.05). Poisson regression test was used to obtain the *p*-values. AIS: acute ischaemic stroke; IQR: interquartile range; TIA: transient ischaemic attack; IHD: ischaemic heart disease; AF: atrial fibrillation; VHD: valvular heart disease; PVD: peripheral vascular disease. The bolded represents a significant difference in the IRR.

## Data Availability

Air quality and meteorological data were retrieved from the publicly available National Environment Agency and Meteorological Service Singapore websites, respectively.

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
