# Peer review of "Clustering of Environmental Parameters and the Risk of Acute Ischaemic Stroke"

_ijerph, 2023, doi:10.3390/ijerph20064979_

Round 1

Reviewer 1 Report

The effect of the environment on acute ischaemic stroke is of interest. This paper has attempted answer a difficult question but the retrospective nature of the data has hindered them. 

Specific Points: Was this a study of a database or reanalysis of the data for a different study?

Page 4 line 168: I would reword this and say "The IRR for Malays, Indians, those with a previous history of stroke,/TIA and those without history of hypertension had a similar IRR as those in cluster 1, the higher IRR remained in all other subgroups".

Discussion: the paper would benefit for a comment on the effects on different ethnic groups.

Limitations: I would add that the study is retrospective should be stated. Also the aggregate nature of the stroke diagnosis needs further comment.

Reviewer 2 Report

The clinical problems arising from environmental exposure to variations in climate including air pollution are an increasing problem. This manuscript describes a detailed study of three groups separated on the  basis of exposure to high wind, high rain or high temperature accompanied by an increased pollution standard index.  They report a significant effect in group 3 and suggests some precautionary measures which should be taken to reduce exposure and he potential risk of acute ischaemic stroke. 

The manuscript presentation can be improved by consideration of the following points.

1. the data used is from 2010 to 2015 and this is now 10 years old.  The authors should add a statement that there haven't been any major climate changes in the subsequent 10 years and if they would expect any variation due to climate change?

2. The Results are important for SE Asia - could something be added in the Introduction comparing data available in SE Asia with that for other major world sectors e. g. Europe and USA?

Line 141 Figure 1 in another paper is referred to - this should be discussed in the Introduction or any relevant information included in this manuscript.  The experimental data in the current paper should "stand-alone" and not rely on data in another paper.

3. Ln 139 to 148 - this could be expanded to make the findings clearer to the non- expert reader. 

4. table 1 This is confusing as the number of males is given and not females is this correct?

Ln 164 "rations" is this correct? 

Ln 191 "were contrasting to ours" should be "contrasted"?

References  the year is omitted from three of the references.
